# miR-129-5p and miR-130a-3p Regulate VEGFR-2 Expression in Sensory and Motor Neurons during Development

**DOI:** 10.3390/ijms21113839

**Published:** 2020-05-28

**Authors:** Kevin Glaesel, Caroline May, Katrin Marcus, Veronika Matschke, Carsten Theiss, Verena Theis

**Affiliations:** 1Department of Cytology, Institute of Anatomy, Ruhr University Bochum, 44780 Bochum, Germany; kevin.glaesel@rub.de (K.G.); veronika.matschke@rub.de (V.M.); verena.theis@rub.de (V.T.); 2Medical Proteom-Center, Ruhr University Bochum, 44780 Bochum, NRW, Germany; caroline.may@rub.de (C.M.); katrin.marcus@rub.de (K.M.)

**Keywords:** microRNA, motor neurons, nervous system, neuronal maturation, sensory neurons, VEGFR-2

## Abstract

The wide-ranging influence of vascular endothelial growth factor (VEGF) within the central (CNS) and peripheral nervous system (PNS), for example through effects on axonal growth or neuronal cell survival, is mainly mediated by VEGF receptor 2 (VEGFR-2). However, the regulation of VEGFR-2 expression during development is not yet well understood. As microRNAs are considered to be key players during neuronal maturation and regenerative processes, we identified the two microRNAs (miRNAs)—miR-129-5p and miR-130a-3p—that may have an impact on VEGFR-2 expression in young and mature sensory and lower motor neurons. The expression level of VEGFR-2 was analyzed by using in situ hybridization, RT-qPCR, Western blot, and immunohistochemistry in developing rats. microRNAs were validated within the spinal cord and dorsal root ganglia. To unveil the molecular impact of our candidate microRNAs, dissociated cell cultures of sensory and lower motor neurons were transfected with mimics and inhibitors. We depicted age-dependent VEGFR-2 expression in sensory and lower motor neurons. In detail, in lower motor neurons, VEGFR-2 expression was significantly reduced during maturation, in conjunction with an increased level of miR-129-5p. In sensory dorsal root ganglia, VEGFR-2 expression increased during maturation and was accompanied by an overexpression of miR-130a-3p. In a second step, the functional significance of these microRNAs with respect to VEGFR-2 expression was proven. Whereas miR-129-5p seems to decrease VEGFR-2 expression in a direct manner in the CNS, miR-130a-3p might indirectly control VEGFR-2 expression in the PNS. A detailed understanding of genetic VEGFR-2 expression control might promote new strategies for the treatment of severe neurological diseases like ischemia or peripheral nerve injury.

## 1. Introduction

Vascular endothelial growth factor (VEGF) was first described in the early 1980s as a tumor-secreted factor which influences vessel permeability [1]. Due to alternative splicing, seven different isoforms are well known: VEGF-A, VEGF-B, VEGF-C, VEGF-D, VEGF-E, VEGF-F, and placental growth factor. At least four different isoforms are noted for VEGF-A—VEGF_121_, VEGF_165_, VEGF_189_, and VEGF_206_—with VEGF-A_165_ being the most biologically active and abundant form [2]. VEGF-A promotes branching, diameter, ingression, and survival of vessels [3], but also shows neurotrophic activity by stimulating axonal growth and cell survival [4,5].

The members of the VEGF family bind to three different receptors: VEGF receptor 1 (VEGFR-1, fms related receptor tyrosine kinase 1 (FLT-1)), VEGFR-2 (kinase insert domain receptor (KDR)), and VEGFR-3 (fms related receptor tyrosine kinase 4 (FLT-4)) [6,7]. These receptors belong to the family of transmembrane tyrosine kinases, which dimerize after ligand binding [8] and phosphorylate their own intracellular domain [5]. VEGFR-1 binds VEGF-A, VEGF-B, and Placental Growth Factor (PIGF) [2] and has significant functions in inflammation [9] and differentiation of hematopoietic stem cells [10]. VEGFR-1, a decoy for VEGF, transmits mitogenic signals in the vascular endothelium and mesenchymal stem cells as well as in the hippocampus and the cerebral motor cortex [2,6,11,12,13]. VEGFR-3 binds VEGF-C and VEGF-D and plays an important role in the regulation of lymphangiogenesis [14]. In the central nervous system (CNS) and especially in the cerebellum, VEGFR-3 is co-expressed with glial fibrillary acidic protein and promotes developmental processes and adult neuronal function in the cerebellum [15].

VEGFR-2 predominately binds VEGF-A and is expressed in the cell surface of most endothelial blood cells [2]. Besides this, the receptor is highly expressed in the neuronal system [16,17]. In astrocytes, VEGFR-2 affects cell proliferation and communication and promotes axonal outgrowth and neuroprotection [4,5,18]. Mediation of VEGFR-2 stimulation by VEGF triggers the activation of cofilin and the Arp2/3 complex, which leads to a fast actin reorganization within the growth cones of dorsal root ganglia (DRG) in the peripheral nervous system (PNS) [18,19]. In the CNS, VEGF/VEGFR-2 signaling increases somato- and denritogenesis in neonatal and juvenile but not mature Purkinje cells (PCs) of the cerebellum [17]. These effects are mainly mediated by VEGFR-2, which is age-dependently expressed during PC maturation and might alter VEGF sensitivity in adult stages [5,17,18].

To date, details on molecular regulation of VEGF/VEGFR-2 signaling are still unknown. At the posttranscriptional and posttranslational level, microRNAs (miRNAs) might have an impact on VEGFR-2 functionality and VEGF sensitivity during neuronal development of the PNS and the CNS [20]. miRNAs are short (19–25 nucleotides), single-stranded, and highly conserved non-coding RNAs [21,22]. These molecules may inhibit the translation of target mRNAs in functional proteins or tag them for degradation [23]. They are well known for their crucial roles in apoptosis, organ development, and genesis of various diseases like colon and breast cancer, Alzheimer’s disease, and Parkinson’s disease [24,25]. Additionally, miRNAs are supposed to play a key role in the development of the nervous system. Piezcora et al. (2017) [26] performed miRNA profiling and revealed 27 miRNAs which were age-related and expressed in the CNS. Based on these data and an intensive PubMed literature research we assumed that two miRNAs—miR-129-5p and miR-130a-3p—might regulate VEGFR-2 functionality in sensory and lower motor neurons. miR-129-5p, a mature form of miR-129-1, and miR-130a-3p have already been described as posttranscriptional regulation factors of VEGFR-2 expression, albeit in endothelial cells [27,28,29]. Soufi-Zomorrod et al. (2016) [27] validated VEGFR-2 as a direct target of miR-129-1 using bioinformatic algorithms and a luciferase reporter assay. miR-129 binds in the 3′ untranslated region (3′ UTR) of VEGFR-2 on mRNA as well as at the protein level. In case of miR-130a, Mujahid et al. (2013) [28] detected a change of VEGFR-2 expression after manipulating miR-130a in fetal lung mouse organ cultures, accompanied by a significant change of *Homebox A5 (Hoxa5)*. To date no direct interaction between miR-130a and VEGFR-2 has been described in the literature or in miRNA databases. Nevertheless, based on the interaction between miR-130a and *Hoxa5* that regulates VEGFR-2, we ascribe a substantial role to miR-130a in regulation and regeneration of VEGFR-2 expression control.

## 2. Results

In this study, we examined the molecular regulation of VEGFR-2 in sensory DRG neurons and lower motor neurons of the spinal cord. Therefore, we analyzed the expression pattern of VEGFR-2 at a posttranscriptional and posttranslational level by using RT-qPCR, immunohistochemistry, in situ hybridization, and Western blotting. Additionally, miR-129-5p and miR-130a-3p were validated at p10 and p30 in sensory and lower motor neurons. The functional influence of these miRNAs with respect to VEGFR-2 expression was proven by transfections of mimics and inhibitors in neuronal cell cultures.

### 2.1. VEGFR-2 Expression in Immature and Mature Sensory Neurons in Drgs

Using in situ hybridization it was shown that VEGFR-2 mRNA *(Kdr)* was expressed in DRG neurons at p30 (Figure 1A, B). In p10 no signal could be detected. Additionally, qPCR data revealed a significant increase in *Kdr* expression by about 24.2% during maturation from p10 (0.783 ± 0.373) to p30 (1.025 ± 0.227) (*n* = 6; ** *p* = 0.0019) (Figure 1C). In addition, the expression of the VEGFR-2 protein within DRG was proven by Western blot (Figure 1D). Additional immunohistochemical analysis exhibited a clear co-staining of VEGFR-2 and anti-phosphorylate neurofilament-H in immature (p10) as well as mature (p30) sensory DRG neurons (Figure 1E, F).

### 2.2. VEGFR-2 Expression in Immature and Mature Motor Neurons of the Spinal Cord

With aid of in situ hybridization, *Kdr* was detectable in immature (p10) and mature (p30) motor neurons in the spinal cord (Figure 2A,B). Additive qPCR disclosed a significant threefold decrease in *Kdr* expression from p10 (3.22 ± 1.071) to p30 (1.033 ± 0.300) (*n* = 6; *** *p* < 0.0001) (Figure 2C). At the protein level, Western blot (Figure 2D) and immunohistochemistry (Figure 2E, F) showed a qualitative VEGFR-2 expression in the lower motor neurons in both age stages.

### 2.3. Age-Dependent MiRNA Expression in Sensory Neurons

To gain deeper insights into the molecular regulation of *Kdr* expression, the effects of miR-129-5p and miR-130a-3p were analyzed by in situ hybridization and RT-qPCR in sensory neurons. This showed that miR-129-5p was not expressed in young or adult sensory neurons of DRG. Concerning miR-130a-3p in sensory DRG neurons, no signal was detectable by in situ hybridization at p10 and p30 (Figure 3A); however, using RT-qPCR the expression of miR-130a-3p in DRG could be confirmed (*n* = 6). Between p10 (1.121 ± 0.587) and p30 (1.116 ± 0.633) the relative amount of miR-130a-3p expression showed non-significant changes (Figure 3B) whereas former studies revealed an age-dependent expression of miR-130a-3p between p30 and p60 (unpublished data of the Department of Cytology).

### 2.4. Age-Dependent miRNA Expression in Lower Motor Neurons

Similar to the work on sensory neurons, we analyzed the expression of both miRNAs in lower motor neurons. There was no differential expression of miR-130a-3p detectable in the spinal cord. With aid of in situ hybridization, miR-129-5p could not be detected in motor neurons at p10 (not shown) but was clearly observable at p30 (Figure 3C). Nevertheless, the quantification of the relative amounts of miR-129-5p revealed a low expression in immature neurons at p10 (0.597 ±0.303), and a significant increase in miR-129-5p expression levels until p30 (1.281 ± 0.996) (*n* = 6) (Figure 3D).

### 2.5. Impact of Mir-130a-3p in Sensory Neurons

To further investigate the impact of miR-130a-3p on the expression of *Kdr*, primary sensory DRG neurons were transfected with miR-130a-3p mimics and inhibitors. To ensure that the effects of mimics and inhibitors were due to their specificity, three controls (untransfected cells, MOCK transfected cells, and a scrambled negative control) were used. Compared to untransfected cells, the application of the transfection reagent and the negative control (NC) did not reveal significant differences in *Kdr* expression (Figure 4A). In contrast, transfection of double-stranded miR-130a-3p mimics led to a significant overexpression of the corresponding miRNA (*n* = 6; ** *p* = 0.005) (Figure 4B). Single-stranded inhibitors of miR-130a-3p efficiently reduced the amount of the corresponding miRNA (*n* = 6; *** *p* < 0.0001) (Figure 4B). With respect to *Kdr* expression, miR-130a-3p mimics did not significantly alter the expression level (0.901 ± 0.260), while the transfection of miR-130a-3p inhibitors significantly decreased the amount of *Kdr* expression (0.783 ± 0.196; *n* = 6; *** *p* < 0.0001) (Figure 4C). Additionally, we analyzed the expression level of *Hoxa5* after transfection with miR-130a-3p mimics and inhibitors. In DRG, miR-130a-3p inhibitors slightly increased *Hoxa5* (1.188 ± 0.394)*,* whereas mimics seemed to reduce *Hoxa5* expression (1.279 ± 0.481) (Figure 4D).

### 2.6. Impact of Mir-129a-5p in Lower Motor Neurons

To analyze the impact of miR-129-5p on the expression of *Kdr* in motor neurons of the spinal cord, these cells were also transfected with specific mimics and inhibitors. As compared to the sensory neurons, the MOCK control and NC had no effect on *Kdr* expression in these lower motor neurons (Figure 5A). To unveil the impact of transfection of miR-129a-5p mimics and inhibitors, RT-qPCR was used to quantify the expression levels of these miRNAs in these primary motor neurons. After transfection of miR-129-5p mimics, the amount of corresponding miRNAs was significantly increased (miR-129-5p: *n* = 6; *** *p* < 0.0001). In contrast, miR-129a-5p inhibitors did not reduce the amount of corresponding miRNA (*n* = 6) (Figure 5B). Mimics of miR-129-5p (** *p* = 0.0013; *n* = 6) significantly decreased *Kdr* expression (0.868 ± 0.201), whereas the impact of miR-129-5p inhibitors was not significant (1.03 ± 0.228) (Figure 5C).

## 3. Discussion

Since the initial description of VEGFR-2 as a key player in angiogenesis and motility of endothelial cells and cancer development [30], the understanding of the neuronal functions has expanded rapidly in recent years [5,31]. There is also increasing scientific and therapeutic interest in VEGF and its corresponding receptors as well as in the differential expression of VEGFR-2 in the PNS [31] and CNS [17]. However, the regulatory mechanisms of its expression are still poorly understood. Therefore, we analyzed miR-129-5p and miR-130a-3p expression in sensory and lower motor neurons to unveil their plausible direct and/or indirect influence on *Kdr* expression during neuronal development. By demonstrating that miR-129-5p and miR-130a-3p expression is linked to VEGFR-2 mRNA expression, it can be assumed that these miRNAs are critical players in VEGFR-2 functionality during development. We therefore characterized VEGFR-2 expression at the mRNA and protein levels. The expression of miR-129-5p and miR-130a-3p in these neurons was then checked using RT-qPCR and in situ hybridization. The possible influence of miR-129-5p and miR-130a-3p on VEGFR-2 expression was tested by transfection of dissociated cell cultures from sensory and lower motor neurons.

### 3.1. Age-Dependent VEGFR-2 Expression in Sensory DRG Neurons and Spinal Cord Motor Neurons

Beside its major impact on angiogenesis, VEGF also shows neuroprotective and neuroregenerative abilities in the nervous system. In the PNS, VEGF promotes neurite outgrowth and accelerates the sensory recovery of injured peripheral nerves in the avascular cornea [32]. In the spinal cord of patients with autoimmune encephalomyelitis, VEGF promotes cell survival in lower motor neurons of the spinal cord from neurodegeneration via VEGFR-2 activation [33]. However, to date the regulatory mechanisms of VEGFR-2 activation mediating neuroprotection in lower motor neurons and the closely located sensory neurons have not been clearly analyzed. Therefore, we initially characterized *Kdr* expression at the mRNA and protein level (VEGFR-2) within sensory and lower motor neurons. It could be shown that *Kdr* expression followed age-dependent regulation processes in both systems. Whereas in matured sensory neurons a significant increase in *Kdr* was detected, lower motor neurons displayed a higher *Kdr* expression at p10 compared to p30. We found expression of VEGFR-2 at the protein level via immunohistochemistry and Western blot in DRG and motor neurons in both age groups.

In 2001 Sondell et al. described an age-dependent regulation of VEGFR-2 activity in DRG using immunohistochemistry [34]. In line with our results, it became clear that VEGFR-2 was an essential factor during development. Expanding the range of methods by quantitative techniques such as RT-qPCR gives a more detailed impression of VEGFR-2 expression in the nervous system, as the current literature demonstrates. In neonatal PCs for example, VEGFR-2 expression is significantly upregulated, whereas in mature stages the amount of VEGFR-2 is decreased [17]. Compatible with these results, lower motor neurons, as a part of the motor system, showed a significant higher amount of *VEGFR-2* at the neonatal age than at matured stages at the mRNA and protein level. Similarly to our study design, previous studies did not differentiate between the sexes. VEGF is known to be regulated in angiogenesis by hormones such as estrogen [35]. However, there is currently no evidence of gender-specific differences in VEGF and VEGFR expression during angiogenesis or neurogenesis. Reynders et al. (2018) showed, for example, that gender has no influence on VEGFR-2 expression in lung cancer patients [36]. Thus, the presented data emphasize that VEGFR-2 plays a decisive role in the entire nervous system, albeit with diverging characteristics. In the CNS VEGFR-2 might control early developmental processes, whereby in the PNS neuronal functionality is maintained by VEGFR-2 also in adults. We assume that posttranslational regulators like miRNAs are involved in these tissue-specific alterations [37].

### 3.2. miRNA Expression in DRG Neurons and Spinal Cord Motor Neurons

miRNAs are key regulators of multiple biological processes like differentiation, metabolism, proliferation, tumorigenesis, and neurodevelopment [38,39,40]. To date the modulation of VEGFR-2 expression in the nervous system is poorly understood. Only in endothelial cells have VEGFR-2/miRNA interactions been described so far [41].

To unveil direct or indirect bonds of miR-129-5p and miR-130a-3p with VEGFR-2, we validated their expression in sensory DRG neurons and motor neurons in the spinal cord. In the present investigation, miR-130a-3p was only detected in sensory DRG, whereas miR-129-5p expression was limited to motor neurons, as confirmed by previous studies [29,42,43]. In DRG neurons only low amounts of miR-130a-3p could be detected. Using in situ hybridization it was not possible to detect miR-130a-3p within these sensory neurons. Nevertheless, the use of RT-qPCR confirmed a constant miR-130a-3p expression at p10 up to p30, accompanied by a significant increase in *Kdr* expression.

In spinal motor neurons, miR-129-5p could be identified by using in situ hybridization. miR-129-5p showed an increased expression in mature motor neurons, whereas *Kdr* expression was significantly decreased at this stage. These data are in line with those of Pieczora et al., postulating an age-dependent expression of miR-129-5p and miR-130a-3p in PCs of the cerebellum also [26]. In these cells miR-130a-3p was downregulated from p9 to p30, whereas miR-129-5p expression was increased in adult rat PCs.

### 3.3. The Influence of miRNA on Kdr Expression

During recent years several miRNAs have been discovered to direct targeting VEGF, e.g., in gliomas or hepatocellular carcinoma cells [44,45]. For example, enforced expression of miR-24 in U251 glioma cells seems to promote the cell viability and angiogenesis in human umbilical vein endothelial cells [44]. In hepatocellular carcinoma miR-205, which might suppress VEGF synthesis to prevent cell growth and metastasis, is downregulated [45]. The modulation of VEGFR-2 expression via miRNAs is described in several systems [46,47]. miR-16, for example, affects the proliferation and angiogenesis of pituitary cancer via VEGFR-2/p38/ nuclear factor ’kappa-light-chain-enhancer’ of activated B-cells (NF-κB) signaling [47]. In human clear cell renal cell carcinoma cells VEGFR-2 is directly targeted by miR-497, which might be a potential strategy to treat renal cell carcinoma [48]. Besides their role in cancer, miRNAs targeting VEGF and its corresponding receptors are discussed with respect to the CNS and PNS [20]. To evaluate the potential influence of miRNAs on *Kdr* expression in the nervous system, we analyzed the impact of miR-129-5p and miR-130a-3p on miRNA and mRNA-level in neuronal cell cultures.

miR-129-5p is an alternative splicing product of miR-129-1 and directly targets *Kdr* [27,49]. The overexpression of miR-129-5p revealed a negative effect on *Kdr* expression, whereas the transfection of miRNA inhibitors had no effect at the miRNA level or the mRNA level. miRNA activity varies among different cell types. Here 30 nM mimics and inhibitors were used based on the work of Salinas-Vera et al. [50]. To unveil the neuroprotective effects of miR-34a in Parkinson’s disease, 30 nM specific inhibitors were transfected into the neuroblastoma cell line SH-SY5Y, resulting in restored cell viability after neuronal damage [51]. In the primary midbrain neurons even lower concentrations of inhibitors have a significant impact on the survival of dopaminergic neurons [52]. However, there are some publications that show higher concentrations of inhibitors compared to mimics in primary Human Umbilical Vein Endothelial Cells (HUVECs) and prostate cancer cells [53,54]. For example, application of 150 nM miR-200a inhibitors efficiently blocked their target miRNA [55]. In case of miR-129-1 Soufi-Zomorrod et al. (2016) revealed a downregulation of VEGFR-2 on mRNA and protein level by upregulation of miR-129-1 in HUVECs using 150 nM inhibitors [27]. In line with this our results indicate that miR-129-5p is capable of regulating VEGFR-2 expression directly in the CNS.

In primary sensory neurons the transfection of mimics and inhibitors led to significant changes in miR-130a-3p levels. However, with regard to *Kdr* and *Hoxa5*, we could detect only insignificant changes in mRNA levels. Solely the transfection with the specific inhibitor resulted in significantly decreased *Kdr* expression. Nonetheless, these data suggest a multi-staged *Kdr* regulation via miR-130a-3p and *Hoxa5* in DRG neurons. The efficient upregulation of *Hoxa5* via miR-130a-3p inhibitors might compete with VEGFR-2 expression in neuronal cells, as described in endothelial cells [28]. The use of 30 nM synthetic miRNA constructs did not lead to these efficient regulation mechanisms in DRG neurons, although 30 nM or even lower concentrations were successfully applied in other neuronal cell cultures [52,56]. Therefore, the final concentration of miR-130a-3p mimics and inhibitors should be increased in further experiments [55].

miR-130a-3p, a c-Myc-responsive miRNA, plays an important role during angiogenesis, and increases VEGFR-2 expression without targeting the receptor directly [28,57]. One interesting candidate seems to be *Hoxa5*, which is directly regulated by miR-130a-3p [58]. *Hoxa5* is known for its important role during embryonic and fetal development of the CNS as well as for an anti-angiogenic effect inter alia through reduction of VEGFR-2 [59,60]. In the present study we could demonstrate that miR-130a-3p indirectly modulated *Kdr* expression in primary sensory neurons. *Hoxa5* inhibition might be a key mechanism in the *Kdr* pathway within the PNS. Downregulation of miR-130a-3p leads to a slight but insignificant increase in *Hoxa5* levels in DRG neurons, followed by a significantly decreased VEGFR-2 expression. The disinhibition of *Hoxa5* and *Kdr* by miR-130a-3p might be an important tool in the regulation of VEGFR-2 functionality. Comparable findings were published by Chen et al. (2008), revealing a binding site for miR-130a in the gene of *Hoxa5*. Further, they showed direct downregulation of *Hoxa5* by miR-130a via plasmid transfection and Western blot in HUVECs [58]. In accordance with our theory, in 2005 Rhoads et al. reported on a reduction of VEGFR-2 at the mRNA and protein level in cells expressing *Hoxa5* [59]. Further work by Mujahid et al. (2013) and Silfa-Mazara (2014) showed interactions between *Hoxa5* and VEGFR-2, whereby the amount of VEGFR-2-positive cells in a fetal lung mouse model was altered by miRNA influence, followed by a reduction in the formation of vascular plexuses around the terminal airways. These findings could be associated with the observed changes in *Hoxa5* localization [28]. Silfa-Mazara et al. (2014) showed explicit changes in airway branching, endothelial cell organization, and VEGFR-2 staining in fetal mouse lung organ cultures after *Hoxa5* induction [59].

### 3.4. Mir-129-5p and Mir-130a-3p could Be Therapeutic Targets in Neurological Diseases

As miRNAs are implicated in the pathogenesis of several neuronal injuries and neurodegenerative diseases, they could possibly be qualified as new therapeutic targets, particularly in the nervous system [61,62]. A therapy using mimics and inhibitors that regulate the functionality of their targets might increase neuronal survival, reduce cognitive impairment, or reduce cell apoptosis e.g., in Parkinson’s or Alzheimer’s disease [52,63,64]. In the present study, investigated miRNAs showed a direct or indirect impact on VEGFR-2 expression, resulting in molecular modulation of the VEGF/VEGFR-2-signaling pathway. Further experiments should investigate whether these miRNAs modulate VEGFR-2 expression in different stages of neurodevelopment or how they are able to promote neuro-regeneration. Current studies strengthen this theory, as miR-26a mimics for example were transfected into brain tissue after cerebral ischemic and were found to promote angiogenesis [65]. Additionally, the overexpression of miR-330 seems to be a useful tool to reduce amyloid ß protein and alleviates mitochondrial dysfunction in an Alzheimer’s disease mouse model [66].

## 4. Material and Methods

### 4.1. Animals and Surgical Procedures

All procedures were conducted under established standards of the German and European legislation. The use of vertebrate animals for scientific purposes in Germany is regulated by the Animla Welfare Act and the institutional Animal Welfare Officer.

Spinal cord and dorsal root ganglia from Wistar rats of postnatal day 10 (p10) and day 30 (p30) were obtained under RNAse-free conditions as previously described by Pieczora et al. 2017 [26]. Working surfaces were cleaned with NaOH–EDTA dissolved in diethylpyrocarbonate (DEPC)-treated (D5758-50ML, Merck, Darmstadt, Germany) water (0.1 M NaOH, 1 mM EDTA (EDS-1kg, Sigma-Aldrich, Merck, Darmstadt, Germany)) before surgical procedures. The operation tools were baked for 4 h at 240 °C. Rats were decapitated after anesthesia with chloroform. The cervical spinal cord and the DRG were dissected and were finally snap frozen using liquid nitrogen and stored at –80 °C.

### 4.2. Immunohistochemistry

The tissues were fixed with 4% paraformaldehyde in Phosphate buffered saline (PBS) for 24 h and washed briefly before paraffin embedding was performed. Then, 10-µm-thick slides were incubated in citrate buffer for 20 min to expose the proteins. After permeabilization with 0.1% Triton (T8532; Sigma-Aldrich, Merck, Darmstadt, Germany) (3 × 5 min), unspecific binding sites were blocked with goat serum (1:50 in PBS) for 30 min. Slices were incubated with primary antibodies dissolved in PBS at 4 °C overnight. The following primary antibodies were used: anti-phosphorylated neurofilament H (1:200, mouse; NA1540, Biotrend, Cologne, Germany) and anti-VEGFR-2 (1:500, goat; V1014, Sigma-Aldrich, Merck, Darmstadt, Germany). After intensive washing with PBS three times, slices were incubated with secondary antibodies such as anti-goat Immunoglobuline G (IgG) Tetramethylrhodamin (TRITC) (1:1000; AF568, Molecular Probes, Thermo Fisher Scientific, Waltham, MA, USA) and anti-mouse IgG FITC (1:200; AF488, Molecular Probes, Thermo Fisher Scientific, Waltham, MA, USA) at room temperature for 2 h. After washing with PBS, nuclear staining was performed by incubation with bisBenzimide H 33342 trihydrochloride (1:1000; HOECHST, B2261, Sigma-Aldrich, Merck, Darmstadt, Germany) for 20 min. Finally, samples were rinsed in PBS and covered with a coverslip in mounting medium (S3023, Dako, F6937, Fluoroshield, Sigma-Aldrich, Merck, Darmstadt, Germany). Stained slices were analyzed with a Zeiss Axiovert 100 M Confocal Microscope (#530258, Jena, Germany).

### 4.3. Western Blot

Western blots were performed on homogenates of rat DRG and spinal cord extracts (p10 and p30). In short, tissue was homogenized in Radioimmunoprecipitation assay buffer (RIPA) buffer (0.5M Tris-HCl (T5941, Sigma-Aldrich, Merck, Darmstadt, Germany), pH 7.4, 1.5M NaCl, 2.5% deoxycholic acid, 10% NP-40, 10 mM EDTA (EDS-1kg, Sigma-Aldrich, Merck, Darmstadt, Germany)). Subsequently, glass beads (A555.1, ROTH, Karlsruhe, Germany) were added to lyse the tissue in an ultrasonic bath (amplitude: 90, cycle 0.5; 4 × 1 min). To separate the tissue from the beads, the solution was centrifuged for 10 min at 4 °C (16,000 *g*). The supernatant was centrifuged again for 15 min at 4 °C (16,000 *g*). After this, protein concentration was measured using a Bradford assay (500-0006, BIO-RAD, Hercules, CA, USA). For 1D gel electrophoresis, 50 µg of protein was used. Then, 10% SDS (74255-250G, Sigma Aldrich, Merck, Darmstadt, Germany), 2M DTT (A1101.0025, AppliChem, Darmstadt, Germany), 4× LDS (MPSB-10ml, Millipore, Merck, Darmstadt, Germany), and H_2_O were added to the samples. Then, the sample was heated to 95 °C for 10 min and transferred to an 4%–12% Bis–Tris gel (NP0323BOX, Thermo Fisher Scientific, Waltham, MA, USA) for 15 min with 50 V and for 60 min with 180 V. After blotting, the membrane was blocked for 2 h with starting block (37578, Thermo Fisher Scientific, Waltham, MA, USA), and then incubated overnight with mouse anti-VEGFR-2 antibody (1:500; V3003, Sigma Aldrich, Merck, Darmstadt, Germany) diluted in starting block and TBS. For glycerin aldehyde 3-phosphate dehydrogenase staining, the antibody (1:5000, rabbit; GTX627408, GeneTex, Irvine, CA, USA) was also diluted in starting block and TBS. After three washing steps in TBS for 10 minutes, secondary antibodies to detect VEGFR-2 (1:150,000, rabbit; IRDye 800, abcam, Cambridge, UK) and glycerin aldehyde 3-phosphate dehydrogenase (GAPDH) (1:15,000, rabbit; IRDye 680, abcam, Cambridge, UK) were incubated for 1 h (VEGFR-2), respectively, for 2 h (GAPDH). Both secondary antibodies were diluted in starting block and TBS, finished by three washing steps in TBS for 10 min.

### 4.4. Cryosections for in Situ Hybridization

After isolation of the DRG and spinal cord the tissue was frozen at –50 °C using isopentane cooled by liquid nitrogen in a small amount of freezing medium (14020108926, Leica, Wetzlar, Germany). The 12-μm sections were sliced with a cryostat (CryoStar NX50 Cryostat, Thermo Fisher Scientific, Waltham, MA, USA), mounted on Superfrost Plus slides (J1800AMNZ, Thermo Fisher Scientific, Waltham, MA, USA) and stored at –80 °C.

### 4.5. In Situ Hybridization

In situ hybridization was performed according to the instruction manual “miRCURY LNA^TM^ microRNA ISH optimization Kit” (90010, Exiqon, Vedbaek, Denmark). Cryosections were incubated 15 min 4% PFA at room temperature. After washing in PBS, the sections were incubated with 2 µg/mL proteinase K (microRNA Detction Set, 90004, Exiqon, Vedbaek, Denmark) for 10 min at 37 °C. For hybridization the tissue was incubated with 80 nM of double Digoxigenin (DIG)-LNA^TM^ mRNA probe (VEGFR-2 custom LNA mRNA detection probe, Exiqon, Vedbaek, Denmark) diluted in in situ hybridization buffer (Enz-33808, Enzo Life Sciences, Lausen, Switerland). We used Actin (60 nM) as a positive control and a double-labeled DIG-scrambled probe as a negative control (60 nM). The tissue was incubated for 2 h at 54 °C. Afterwards the tissue was washed once with 5× saline sodium citrate (SSC), twice with 1 × SSC, and 0.2 × SSC for 5 min at hybridization temperature and finally once with 0.2 × SSC for 5 min at room temperature. After washing with PBS for 5 min the slides were treated for 15 min with blocking solution and subsequently incubated with anti-digoxigenin alkaline phosphatase fab fragments (1:800, sheep; 11093274910, Roche, Basel, Switzerland) overnight at 4 °C. On the following day, the slides were washed with PBS (5 × 3 min), followed by NBT-BCIP counterstaining (11697471001, Roche, Basel, Switzerland). The AP substrate was incubated for 5 h in the dark at 30 °C. Finally, the tissue was incubated with 50 μL Nuclear Fast Red (N3020, Sigma-Aldrich, Merck, Darmstadt, Germany) at each slide for 1 min, washed, dehydrated, and fixed with mounting medium. The tissue was stored over night at 4 °C and analyzed by light microscopy (Olympus BX 61) the subsequent day.

MicroRNAs were detected using double DIG-LNA^TM^ microRNA probes (#611350-360, #610790-360, Exiqon, Vedbaek, Denmark). As controls, double DIG-LNA^TM^ U6 snRNA probes and double DIG-LNA^TM^ Scramble-miR probes (both 90010, Exiqon, Vedbaek, Denmark) were used. AP substrate was incubated for 60 min in the dark at room temperature.

### 4.6. mRNA- and Total RNA Extraction, Reverse Transcription, and RT-Qpcr

mRNA was extracted from rat spinal cord and DRG using the ReliaPrep RNA Tissue Miniprep (Z6111, Madison, WI, USA) System according to the manufacturer’s protocol (P. The tissue (20 mg) was incubated in lysis buffer. ReliaPrep^TM^ Minicolumns were equilibrated and prepared according to the manufacturer’s protocol. RNA was eluted using 30 µL RNase-free water and frozen at –80 °C for storage. mRNA reverse transcription was performed using qScript cDNA Synthesis Kit (95047-025, Quantabio, Beverly, MA, USA) according to the manufacturer´s protocol using 100 ng mRNA. To quantify mRNA amounts RT-qPCR was performed using GoTaqR qPCR Master Mix (A6001, Promega, Madison, WI, USA) according to the manufacturer’s protocol. The following primer sequences were used:

*Kdr* (5′-TCC CAG AGT GGT TGG AAA TG-3′, 3′-ACT GAC AGA GGC GAT GAA TG-5′, Microsynth, Balgach, Switzerland), *Hoxa5* (5′- TAG TTC CGT GAG CGA ACA ATT C-3′, 3′- GCTGAGATCCATGCCATTGTAG-5′, Microsynth, Balgach, Switzerland), and GAPDH (5′-ACT CCC ATT CTT CCA CCT TTG-3′, 3′-CCC TGT TGC TGT AGC CAT ATT-5′, Microsynth, Balgach, Switzerland). GAPDH was used as a reference gene.

For miRNA analysis, total RNA containing small and large RNA molecules was extracted from DRG and spinal cord (both 30 mg) as well as from dissociated cell cultures using NucleoSpin miRNA kit (740971.10, Macherey-Nagel, Düren, Germany) according to the manufacturer’s protocol. After primary washing steps, minispin columns were used to extract small RNA molecules. Finally, the RNA was eluted in 50 μL of RNase-free water and stored at –80 °C. cDNA was synthesized using the Universal cDNA synthesis kit II (203301, Exiqon, Vedbaek, Denmark) according to the manufacturer’s protocol using 10 ng RNA. For RT-qPCR using GoTaqR qPCR MasterMix (5 µL), the following primers were applied: hsa-miR-129-5p (204534, Exiqon, Vedbaek, Denmark) and hsa-miR-130a-3p (204658, Exiqon, Vedbaek, Denmark). Synthetic miRNA U6 was used as a reference gene.

According to the manufacturer’s protocol, samples were heated at 95 °C for 10 min, followed by 45 cycles including 10 s at 95 °C and 1 min at 60 °C. RT-qPCR was performed using a CFX Connect^TM^ Real-Time PCR Detection System (Bio-Rad, Hercules, CA, USA). Data were analyzed using the CFX Manager^TM^ Software (version 3.1, Bio-Rad, Hercules, CA, USA).

### 4.7. Cell Culture

Primary sensory neurons were prepared as described previously [67]. In brief, Wistar rat pups at postnatal day 7 were used to obtain DRG. DRG were isolated form the spinal cord under a binocular microscope. After three washing steps with dissociation solution containing Hank’s Balanced Salt Solution HBSS (H8264, Sigma-Aldrich, Merck, Darmstadt, Germany), 2-(4-(2-Hydroxyethyl)-1-piperazinyl)- ethansulfonsäure (HEPES) (A1069,0100, AppliChem, Darmstadt, Germany), and penicillin/streptomycin (P433, Sigma-Aldrich, Merck, Darmstadt, Germany), DRG were incubated in Collagenase II (17101-015, Gibco, Thermo Fisher Scientific, Waltham, MA, USA) solution followed by 2.5% trypsin (15090-046, Gibco, Thermo Fisher Scientific, Waltham, MA, USA) solution for 10 minutes. After further washing steps, the cells were centrifuged twice for five minutes before seeding the cells in a 96-well plate coated with Poly-D-lysin (P7280, Sigma-Aldrich, Merck, Darmstadt, Germany). After 30 minutes, medium containing L-glutamine (G7513, Sigma-Aldrich, Sigma-Aldrich, Merck, Darmstadt, Germany), B-27^®^ supplement (17504044, Thermo Fisher Scientific, Waltham, MA, USA), penicillin/streptomycin (P4333, Sigma-Aldrich, Merck, Darmstadt, Germany), Neurobasal^®^-A medium (10888022, Thermo Fisher Scientific, Waltham, MA, USA) and β-NGF (25 ng/µL) (N0513, Sigma-Aldrich, Merck, Darmstadt, Germany) was changed. Finally, these cultures were placed in an incubator at a temperature of 37 °C for 3 days before transfection.

Motor neuron cell cultures were prepared as previously described by Montoya-Gacharna et al. (2012) [68] and Brewer et al. (2007) [69] via gradient centrifugation. p0 rats were used to obtain the spinal cord. Tissue was incubated with DNase (LS002139, Cell Systems, Troisdorf, Germany), and papain (LS003119, Cell Systems, Troisdorf, Germany) for five minutes at room temperature and 25 min at 30 °C. After centrifugation, cells were resuspended in neural induction medium, containing Hibernate A (A1247501, Thermo Fisher Scientific, Waltham, MA, USA), penicillin/streptomycin (P4333, Sigma-Aldrich, Merck, Darmstadt, Germany), B-27^®^ supplement (17504044, Thermo Fisher Scientific, Waltham, MA, USA) and Glutamax (35050061, Thermo Fisher Scientific, Waltham, MA, USA), with DNase. The suspension was centrifuged for 15 min at 2500 rpm and 10 °C with the gradient. After isolating the third fraction of gradient centrifugation, cells were quantified and seeded in a 96-well plate coated with Poly-D-lysine. After one day of cultivation the cells were illustrated with a Zeiss Axiovert 25 Inverted Phase Contrast microscope (#200301, Jena, Germany).

### 4.8. Transfection

Transfection was performed using Polyplus INTERFERin^®^ in vitro siRNA/miRNA transfection reagent (409-10, Polyplus, Illkirch-Graffenstaden, France), established for the transfection of postmitotic neuronal cell types [70]. miRNA mimics and miRNA inhibitors (30 nM), synthesized by GenePharma (GenePharma, Shanghai, China), were used for transfection [51]. After mixing the duplexes with culture medium, INTERFERin^®^ was added and the mixture was vortexed immediately for 10 s. After incubation for 10 minutes at room temperature to allow INTERFERin^®^/miRNA complexes to form, 50 μL was added to 125 μL of cell suspension in culture medium. The plate was incubated for 30 h at 37 °C before washing and lysing the cells. Untransfected cells, a MOCK group (blank control, no sequence transfected), and a negative control (transfection of a meaningless sequence) served as controls.

### 4.9. Statistical Analysis of RT-qPCR

To receive statistically relevant data of gene expression, six animals of p10 and p30 were used. Fold change of expression was calculated using the 2^-ΔΔ*CT*^ method. The relative expression levels of validated RNAs were compared using an unpaired two-tailed *t*-test. Results for both systems were normalized against p30 and untransfected cells, respectively, for transfection. For statistical analysis, Microsoft Excel Version 16.35 and Graph Pad Prism Version 5.0a were used.

## 5. Conclusion

For the first time, our study reveals the age-dependent expression of VEGFR-2 in DRG neurons and motor neurons in the spinal cord at the mRNA and protein level. Apparently, the expression of two miRNAs—miR-129-5p and miR-130a-3p—is involved in these regulative processes. Interestingly, to date no data have addressed the interaction between VEGFR-2 and our candidate miRNAs in the nervous system. Nevertheless, our study confirms the differential expression of miR-129 and miR-130 that authorized a hypothetic mechanism for each miRNA influencing VEGFR-2 expression. miR-129-5p potentially regulates VEGFR-2 expression by degrading its target directly. miR-130a-3p modulates VEGFR-2 as well, but we assume that the expression is altered through a multi-stage process including *Hoxa5*. *Hoxa5* seems directly to inhibit VEGFR-2, whereas the limitation of *Hoxa5* activity might result in increased VEGFR-2 expression by the mechanism of disinhibition (Figure 6). VEGFR-2 could have a tremendous impact on the development and regeneration of sensory and lower motor neurons.

In the PNS, miR-130a-3 depresses *Hoxa5*, leading to an enhanced level of VEGFR-2 expression. In conclusion, the overexpression of miR-130a-3p might enhance the neuroprotective potential of VEGFR-2 after, e.g., peripheral nerve lesions.

The findings presented here provide initial evidence that miR-129 and miR-130 might play an important role in VEGFR-2 expression control in developing neural cells of the CNS and PNS. To uncover these complex regulatory mechanisms completely, further investigations at the genetic level have to be accomplished. The major challenge will be to identify interaction partners of miR-129 and miR-130 that show an impact on VEGFR-2 functionality

## Figures and Tables

**Figure 1 ijms-21-03839-f001:**
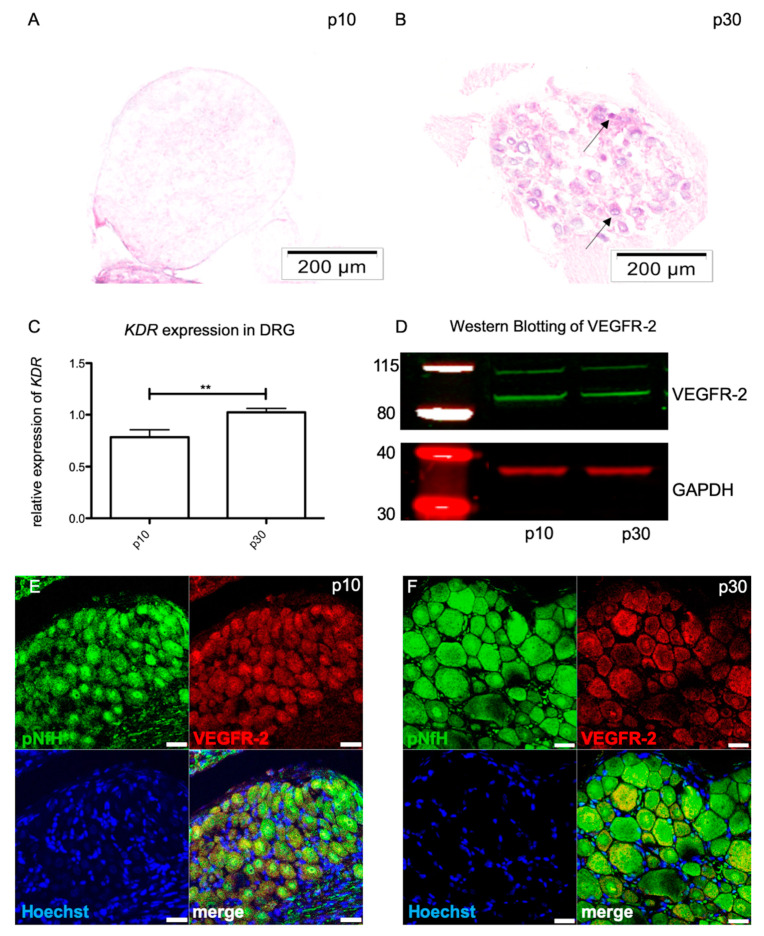
Expression of vascular endothelial growth factor receptor 2 (VEGFR-2) in dorsal root ganglia (DRG). In situ hybridization shows *kinase insert domain receptor* (*Kdr)* expression in DRG neurons at postnatal (p) 10 (**A**) and p30 (**B**). (**C**) RT-qPCR revealed a differential expression of *Kdr* with a significant increase from p10 to p30. The 2^-ΔΔ*CT*^ method was accomplished by using the housekeeping gene *GAPDH* for normalization; data were tested for significance using unpaired *t*-test (** *p* = 0.0019). (**D**) Expression of VEGFR-2 protein (90 kDA) in DRG at p10 and p30. (**E**, **F**) Immunohistochemistry revealed VEGFR-2 expression in perikarya of sensory neurons at p10 (**E**) and p30 (**F**). Scale bars: **A, B**: 200 μm; **E, F**: 20 μm.

**Figure 2 ijms-21-03839-f002:**
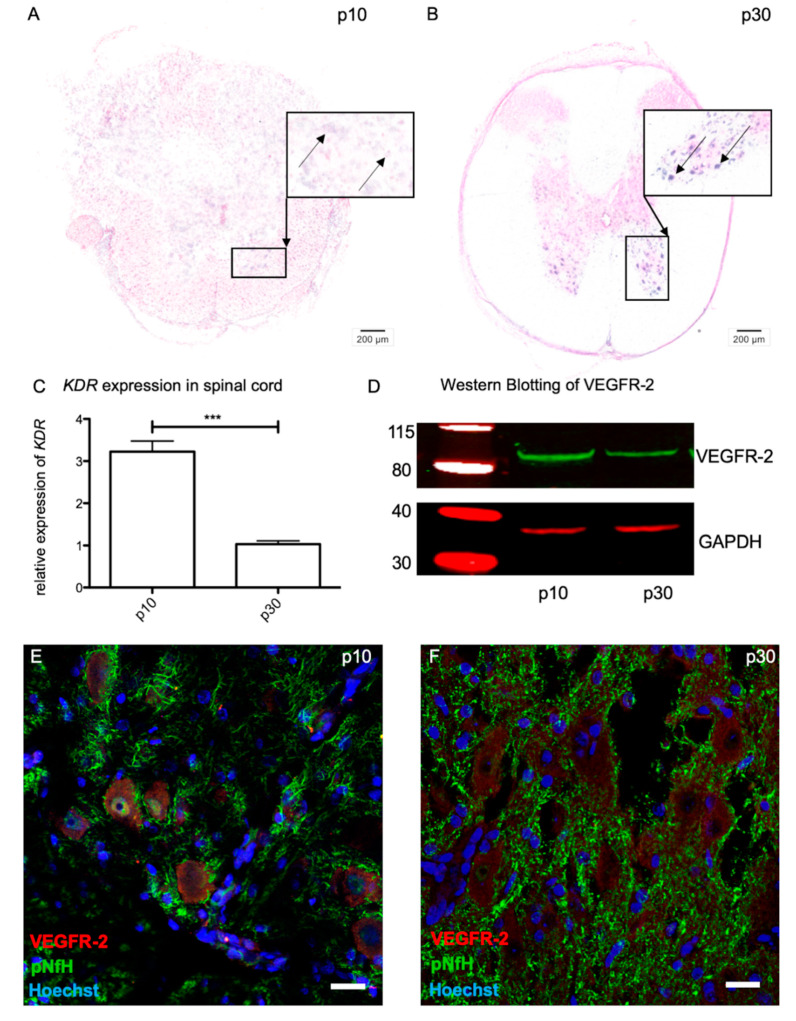
VEGFR-2 expression in the spinal cord. In situ hybridization showing *Kdr* expression in spinal cord motor neurons at p10 (**A**) and p30 (**B**). (**C**) *Kdr* expression was significantly increased at p10 compared to p30. For relative quantification of *Kdr* expression, the 2^-ΔΔ*CT*^ method was accomplished by using the housekeeping gene *GAPDH* for normalization; data were tested for significance using an unpaired *t*-test. Significant differences are indicated by *** *p* <0.0001. (**D**) Expression of VEGFR-2 protein (90 kDA) in lower motor neurons at p10 and p30. (**E**) and (**F**) Verification of VEGFR-2 protein expression in lower motor neurons via immunohistochemistry. Scale bars: **A, B:** 200 μm; **E, F**: 20 μm.

**Figure 3 ijms-21-03839-f003:**
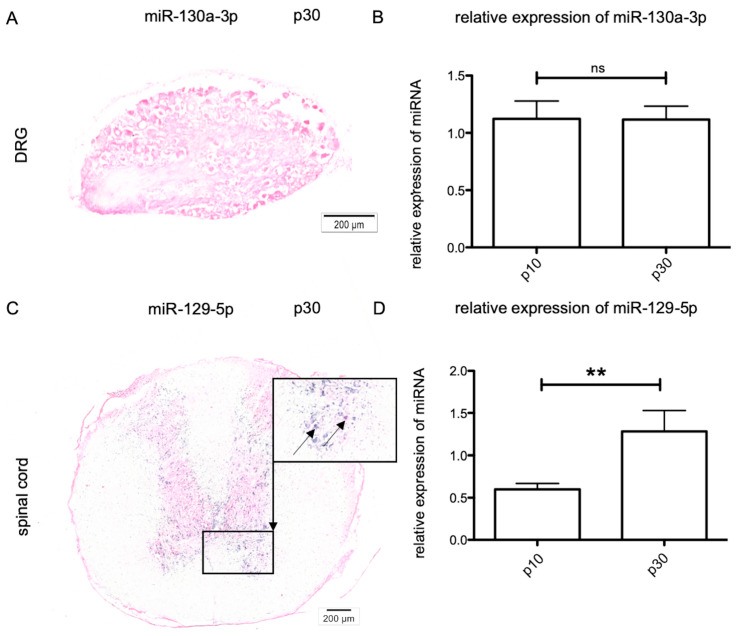
Expression of miRNA (miR)-130a-3p in DRG and miR-129-5p in lower motor neurons at p10 and p30. (**A**) In DRG neurons, miR-130a-3p could not be detected by in situ hybridization; however, miR-130a-3p was detected with aid of RT-qPCR, with non-significant (ns) changes concerning the expression level between p10 and p30 (**B**). (**C**) In situ hybridization disclosed miR-129-5p expression in spinal cord motor neurons, with a significant increase in the relative expression at p30, revealed by RT-qPCR (**D**). For relative quantification of miRNA expression, the 2^-ΔΔ*CT*^ method was accomplished by using the miRNA U6 for normalization. Data were tested for significance using an unpaired *t*-test. ** *p* = 0.0091. Scale bars: **A**, **C**, 200 μm.

**Figure 4 ijms-21-03839-f004:**
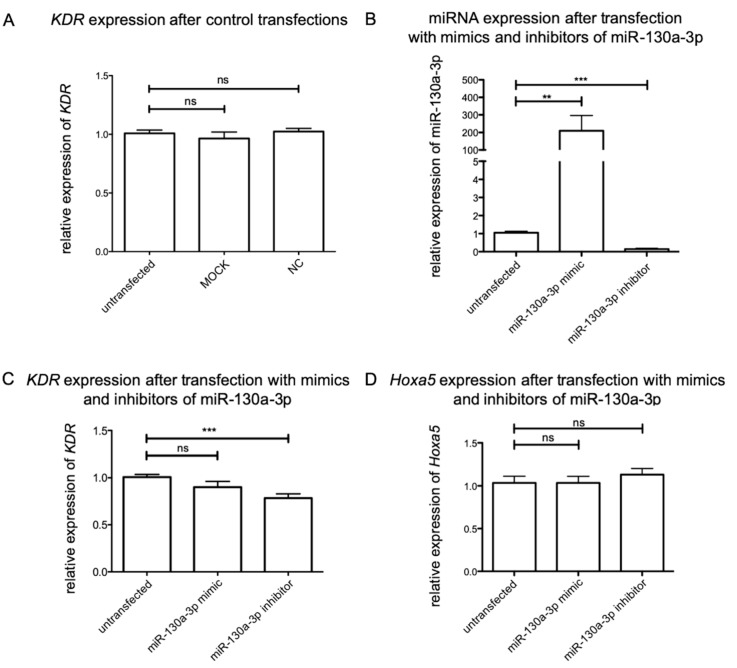
Transfection of DRG with miR-130a-3p mimics and inhibitors. (**A**) Neither the transfection reagent (MOCK) nor the negative control (NC) had a significant impact on *Kdr* expression. (**B**) miR-130a-3p mimics increased the expression of their corresponding miRNA, whereby inhibitors decreased their expression significantly (** *p* = 0.0059; *** *p* < 0.0001). (**C**) miR-130a-3p inhibitors repressed *Kdr* expression efficiently (*** *p* < 0.0001). (**D**) *Homebox A5* (*Hoxa5)* expression was not affected by miR-130a-3p inhibitors or mimics. For relative quantification of mRNA expression, the 2^-ΔΔ*CT*^ method was accomplished by using the housekeeping gene *GAPDH* for normalization; data were tested for significance using an unpaired *t*-test.

**Figure 5 ijms-21-03839-f005:**
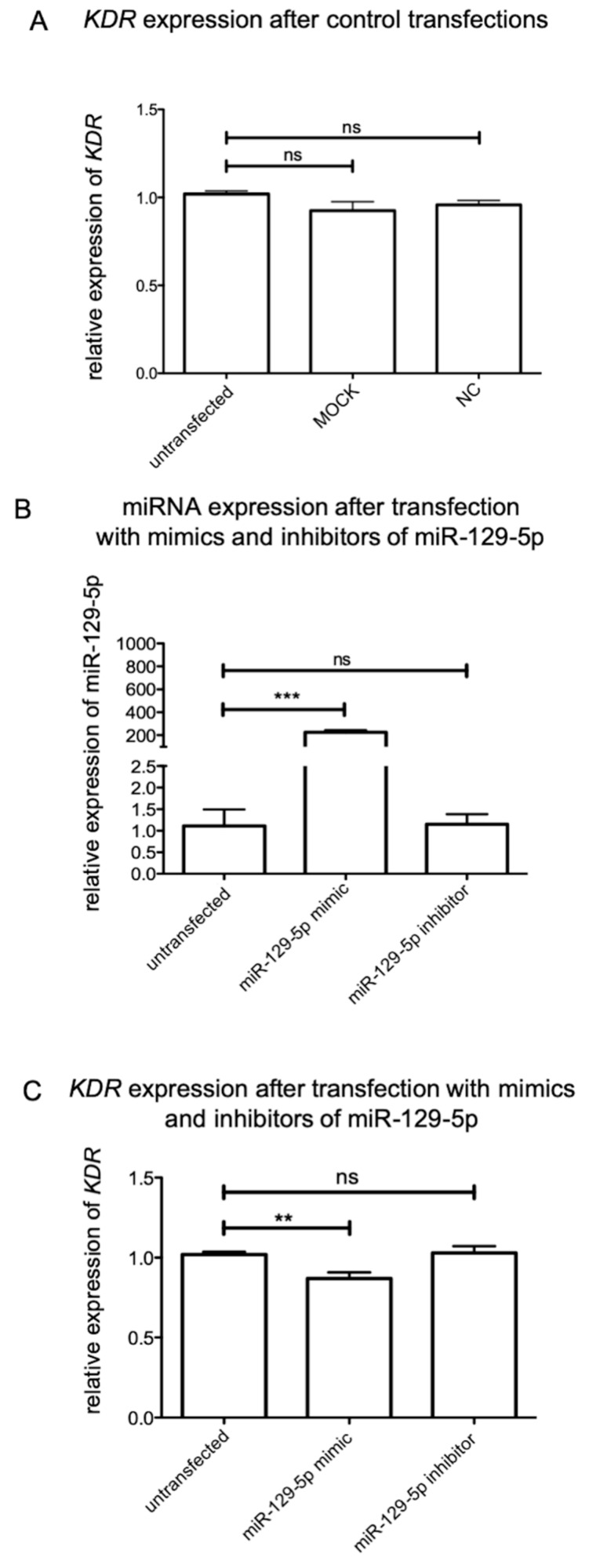
Transfection of motor neurons with miR-129a-5p mimics and inhibitors. (**A**) Transfection with MOCK or NC revealed no significant effect on *Kdr* expression. (**B**) Mimics of miR-129a-5p increased the expression of their corresponding miRNA, whereby inhibitors did not decrease their expression. (*** *p* < 0.0001). (**C**) Additionally, mimics of miR-129-5p (** *p* = 0.0013) significantly decreased *Kdr* expression, whereas the impact of miR-129-5p inhibitors was not significant. For relative quantification of *Kdr* expression, the 2^-ΔΔ*CT*^ method was accomplished by using the housekeeping gene *GAPDH* for normalization; data were tested for significance using an unpaired *t*-test.

**Figure 6 ijms-21-03839-f006:**
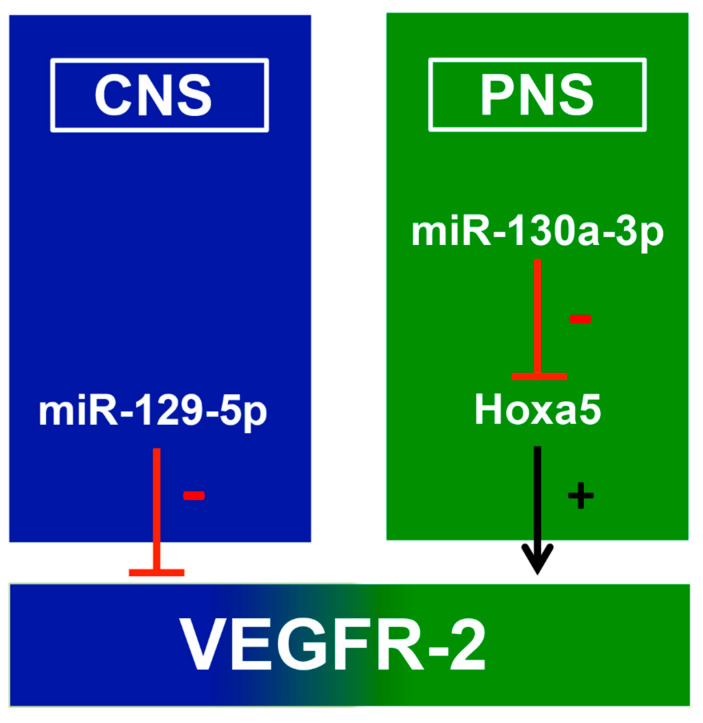
Impact of miR-129-5p and 130a-3p on VEGFR-2-expression in the nervous system. In the central nervous system (CNS), miR-129-5p directly targets VEGFR-2 and decreases its expression (red line). The directed downregulation of miR-129-5p via inhibitors might elevate the amount of functional VEGFR-2 and consequently a powerful therapy to treat nerval regeneration. In the PNS miR-130a-3p the most promising tool to regulate indirectly VEGFR-2 expression. The overexpression of miR-130a-3p might downregulate Hoxa5 levels, followed by an increased VEGFR-2 expression (black arrow).

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
