# Peer review of "miR-129-5p and miR-130a-3p Regulate VEGFR-2 Expression in Sensory and Motor Neurons during Development"

_ijms, 2020, doi:10.3390/ijms21113839_

Round 1
Reviewer 1 Report
The authors did not improved the quality of the manuscript as recommended and therefore did not solve the major issues.
The only experiments investigating the possible effect of miRNA deregulation in neurons are presented in Figure 4 and 5:
a) All the experiments shown are qPCR, the KDR protein expression is not quantified. No effect of miR-130-3p is shown nor in condition of miRNA overexpression nor when the miRNA is inhibited (see Fig.4c and D).
b) The only very mild effect of miR-129-5p (less than 20%) shown in Fig.5C is in condition of miRNA over-expression but no effect is shown when the miRNA is inhibited. Of note, a result obtained by qPCR to be considered significant must be over 2 fold change difference. No data on protein expression level are shown.
On the basis of the data shown in the manuscript the conclusions are not supported by the results presented.
Author Response
We thank you for your effort and all your helpful critical advices which we tried to set up the best way and which help to improve the manuscript.
1.1 We are thankful for your advice. We illustrate a significant impact of miR-130a-3p inhibition on Kdr expression in DRG in Figure 4C. As described in the first revision letter it would be necessary to quantify VEGFR-2 expression on protein level. But as it is mentioned in the paper, Western Blots were performed using tissue from DRG and spinal cord tissue. We had to use high protein amounts (50 µg) to get valid results. Unfortunately, it is not possible to work with such high protein amounts from transfected primary cell cultures containing about 30.000 cells per well. In previous Western Blots we tried to validate positive signals for actin or GAPDH in the cell culture. But because of the very low amounts of protein it was neither possible to detect these high abundant proteins nor the receptor.
In this study we could only validate qualitative VEGFR-2 expression. In further studies the quantitative protein level of VEGFR-2 has to be investigated intensively.
1.2 Thank you for this comment. As we described in the first revision letter we used five animals for every cell culture and transfection. We replicated the transfection six times. Taken together we have a pool of cells from thirty different animals for each transfection of DRG and lower motor neurons. Associated with the 2-ΔΔCT method and the statistical t-test, we assume a result which could detect slight differences in expression levels.
Additionally, our goal was to quantify VEGFR-2 expression via Western Blot but we were forced to stop our work at the laboratory due to the shutdown of our university. Because of the lack of results, we attenuate the analysis of the Western Blot in the corresponding section and only want to show a qualitative examination of VEGFR-2 on protein level.
Reviewer 2 Report
This new version of manuscript is suteable to the publication.
Author Response
We are very grateful to this reviewer for her/his positive comments and feedback.
Reviewer 3 Report
This is a well-designed study that compliments previous studies and reinforces the role of specific microRNAs.
Author Response

(The authors gave the same response as above.)

Reviewer 4 Report
microRNA-129 and microRNA-130 regulate VEGFR-2 expression in sensory and motor neurons during development
In this article, Glaesel et al. use spinal cord and DRG to quantify the expression of Kdr, the gene that codes for Vegfr-2, at two developmental time point p10 and p30. Consistent with the literature, Kdr expression increases with age in the DRG but decrease in the spinal cord. They then show a relationship of 2 miRNAs 130a-3p and 129-5p in sensory DRG and lower motor neurons respectively. Finally, they performed in vitro manipulations to assess the relationship between the miRNAs of interest and the target Kdr and suggest a model of direct and indirect regulation of Kdr in the spinal cord and DRG, respectively.
This paper was well written, the content is very interesting, and appropriate experiments were performed. However, the conclusion are somewhat overdrawn and there are some methodological concerns to consider.
Major concerns:
- The authors found no change in mir-130a-3p expression from P10 to P30 in the DRG but a change to Kdr over time in both the DRG and spinal cord. Nonetheless, they continued with in vitro studies to manipulate the level of mir-130a-3p primary DRG neurons. Forcing an in vitro system can result in changes that are not necessarily biologically relevant. If the data from animals does not suggest a change in the level of this regulator over time, what is the logic in forcing this change in an artificial system? If the authors wish to suggest that mir-130a-3p could be used therapeutically, then what is the point of looking for change across developmental periods, these seem to be two different questions.
- As a follow up to the mir-130a-3p findings, given that its increase is positively related to the levels of Kdr, the authors suggest that it is mediating another transcript Hoxa5, but do not provide any rationale for why they think mir-130-3p might be regulating the level of The authors simply state “One protein that is directly regulated by miR-130a-3p is Hoxa5” without a citation or any evidence. Further, they find that altering the levels of mir-103a-3p does not significantly influence the level of Hoxa5 but somehow conclude that “In the present study we could demonstrate for the first time that miR-130a-3p modulates KDR expression through Hoxa5 inhibition in the PNS”. I don’t agree that there is any evidence for this, there is an indirect relationship between a mir and a transcript that could be mediated by a number of mechanisms. There is no evidence, that the mir-130a-3p regulates Hox5a and there is no evidence that this is the mechanism increasing the level of Kdr. One would expect if this is the mechanism, that increasing the level of the miRNA would have the opposite effect, i.e decrease Hoxa5 and, in turn, increase Kdr. This was not found.
- For miR-129-5p, the inhibitor didn’t work, and the authors speculate that this could be a problem of dosage, it is also possible that the inhibitor was not well designed. Did the authors test more than one construct?
- Given the reliance of the finding on the in vitro experiments, did the authors test the transfection rate to be sure the cells are transfecting well, this may also explain why the attempted inhibition didn’t work.
- Figure 3 needs to be labeled more clearly so the readers understand that the displayed information refers to different regions. Also, it is not clear what the expression level of mir-130-3p is in lower motor neurons and why this data was not included.
- Given the lack of direct evidence, the authors need to tone down their concluding statements.
Minor comments:
- For mice or rats, gene symbols generally are italicised, with only the first letter in uppercase and the remaining letters in lowercase (Kdr). Protein designations are the same as the gene symbol but are not italicised and all are upper case.
- Line 82 has a typo – it reads “Up to now the is now”, it should read “Up to now there is no”.
- Why was there no attempt to quantify the western blots?
- Typically, more than one housekeeping gene should be used to normalize qPCR data.
Author Response
We would like to thank reviewer 4 for her / his efforts and all helpful critical advices. We have tried to contribute to the best possible improvement of the manuscript through linguistic clarifications and additions.
4.1 According to your question we add further results. We hope that these results will make it clearer now. After the insignificant expression of miR-130a-3p and miR-129-5p in DRG we test further ages like p60 to get a deeper insight into the expression trend of both miRNAs in DRG. We could show an age-depended expression for miR-130a-3p between p30 and p60 as shown in the figure below. Because of the interesting prescribed functions and the age-dependent expression of miR-130a-3p between p30 and p60, we suggest that the in vitro studies with the miRNA would be especially interesting.
4.2 We thank the reviewer for this input and modified the corresponding paragraphs. As we described in the discussion, we assume Hoxa5 as a key player in the multi-staged expression mechanisms of miR-130a-3p and VEGFR-2. miR-130a-3p is known for an increasing effect on mRNA/protein-expression. Hoxa5 works normally as an inhibitor of VEGFR-2 expression. To explain the increasing effect of miR-130a-3p on VEGFR-2 expression, we assume a disinhibition mechanism by blocking Hoxa5 through miR-130a-3p.
Page 12, line 298 – 305:
“(…) One interesting candidate on protein level seems to be Hoxa5 that is directly regulated by miR-130a-3p. Hoxa5 is known for its important role during embryonic and fetal development of the CNS [58,59]. In the present study we could demonstrate that miR-130a-3p modulates indirectly Kdr expression in primary sensory neurons. Hoxa5 inhibition might be a key mechanism in the Kdr pathway within the PNS. Downregulation of miR-130a-3p leads to a slight but insignificant increase of Hoxa5 levels in DRG neurons followed by a significantly decreased VEGFR-2 expression. The disinhibition of Hoxa5 and Kdr by miR-130a-3p might be an important tool in the regulation of VEGFR-2 functionality. (…)”
Page 15, line 469 – 475:
“(...) miR-130a-3p modulates VEGFR-2 as well, but we assume that the expression is altered through a multi-staged processing including Hoxa5. Hoxa5 could directly inhibit VEGFR-2 and solely a limitation of Hoxa5 activity could be capable to raise VEGFR-2 expression (Figure 6). (...)“
4.3 Thank you very much for this advice. We used the construct in former studies with different neuronal cells before and simultaneously to this study. The construct worked very well in the other studies, so we suppose that it is a problem of dosage. Additionally, the construct was designed by GenePharma Shanghai, a company often cited in other papers using mimics and inhibitors.
4.4 We are very grateful for your comment on our difficulties to measure the transfection efficiency for miR-129-5p inhibitors. During our literature research we could find some papers with different amounts of mimics and inhibitors each. We discuss these dosage problems for miR-130a-3p on page 11-12 from line 323 – 336. In the end we assume for miR-129-5p as well as for miR-130a-3p a dosage problem and will try different amounts of mimics and inhibitors to solve this difficulty in further studies.
4.5 We are very thankful for this advice and modified the figure. We hope that it is clearer now. The expression level of miR-130a-3p in lower motor neurons was not altered between p10 and p30, so we did not continue the studies with this miRNA in lower motor neurons.
4.6 Thank you very much for your comment to help us to improve our manuscript. As described above we modified corresponding paragraphs.
4.7 Thanks for this comment. We correct the gene symbols and protein designations in the whole paper.
4.8 Thank you very much for this advice. We correct this sentence.
Page 2, line 82:
“(…) Up to now there is no (…)”
4.9 Thanks for this question. We attempted the quantification of the Western Blot subsequently receiving the first revision. The time period between our first Western Blots and the shutdown of our university was too short to reproduce and quantify the results shown in the paper. Because of the sustained shutdown, we could not use our lab in the last months.
4.10 We are very pleased about this comment. According to former studies (e.g.: Herrfurth et al., “Morphological Plasticity of Emerging Purkinje Cells in Response to Exogenous VEGF”, (2017), Frontiers Molecular Neuroscience;Vangeel et al., “Functional expression and pharmalogical modulation of TRPM3 in human sensory neurons”, (2020), British Journal of Pharmacology) we used only one housekeeping gene to normalize qPCR data.
Round 2
Reviewer 1 Report
The authors assume that a 20% difference of expression observed by qPCR is relevant and sufficient to demonstrate that a mifroRNA regulates gene expression; this is not only an overstatement but do not support the conclusion of this manuscript.
The comments made in the previous two revisions have not been addressed.
Author Response
We are very thankful for your critical advices. We used qPCR and statistical analysis to get reliable and valid results. In this study the biological and technical variance were considered by using 30 wistar rats in total for each tissue as well as by repeating the transfection and qPCR six times. Further we strengthened our results with a t-test showing us significant differences between two ages.
In the end we are convinced that our results are reliable and valid enough to detect differences under 20 %.
Reviewer 4 Report
Please keep in mind your finding are correlational, you do not show direct binding of the miRNA to target, for example, a target blocker was not used.
In the case of Hoxa5, you still have not shown evidence that mir130 binds or is predicted to bind to the transcript, and given that any given miRNA has many targets, the effects on Kdr may be mediated by another intermediary.
Please be sure to include this in the limitations.
Author Response
We thank you for your comments to improve our publication. With this in mind, we have conducted further literature research to support our hypothesis. We speculate that miR-130a-3p affects VEGFR-2 in a multistep processing including Hoxa5.
Hoxa5 is known for its anti-angiogenic effect, especially for inhibition of VEGFR-2. Our theory is that miR-130a-3p inhibits Hoxa5 and increases VEGFR-2 through a mechanism of disinhibition. In previous paper versions we could only relate VEGFR-2 and Hoxa5. Due to your very valuable comments we could now find another publication that shows a direct effect of miR-130a-3p on Hoxa5.
With this we can now significantly strengthen our entire theory and show a direct effect of miR-130a-3p, Hoxa5 and VEGFR-2. We have rewritten this part according to their suggestion in the discussion and added two more publications.
Page 12, line 297 – 316:
“(…) miR-130a-3p, a c-Myc responsive miRNA, plays an important role during angiogenesis, which increases VEGFR-2 expression without targeting the receptor directly [28,56]. One interesting candidate seems to be Hoxa5 that is directly regulated by miR-130a-3p [57]. Hoxa5 is known for its important role during embryonic and fetal development of the CNS as well as for an anti-angiogenic effect inter alia through reduction of VEGFR-2 [58,59]. In the present study we could demonstrate that miR-130a-3p indirectly modulates Kdr expression in primary sensory neurons. Hoxa5 inhibition might be a key mechanism in the Kdr pathway within the PNS. Downregulation of miR-130a-3p leads to a slight but insignificant increase of Hoxa5 levels in DRG neurons, followed by a significantly decreased VEGFR-2 expression. The disinhibition of Hoxa5 and Kdr by miR-130a-3p might be an important tool in the regulation of VEGFR-2 functionality. Comparable findings were published by Chen et al. (2008), revealing a binding site for miR-130a in the gene of Hoxa5. Further they showed direct downregulation of Hoxa5 by miR-130a via plasmid transfection and Western Blot in HUVEC [57]. According to our theory, Rhoads et al. reported already in 2005 about a reduction of VEGFR-2 on mRNA and protein level in cells expressing Hoxa5 [58]. Further work by Mujahid et al (2013) and Silfa-Mazara (2014) showed interactions between Hoxa5 and VEGFR-2, whereby the amount of VEGFR-2-positive cells in a fetal lung mouse model was altered by miRNA influence, followed by a reduction in the formation of vascular plexuses around the terminal airways. These findings could be associated with the observed changes in Hoxa5 localization [28]. Silfa-Mazara et al (2014) showed explicit changes in airway branching, endothelial cell organisation and VEGFR-2 staining in fetal mouse lung organ cultures after Hoxa5 induction [59]. (…)”
This manuscript is a resubmission of an earlier submission. The following is a list of the peer review reports and author responses from that submission.
Round 1
Reviewer 1 Report
Title: microRNA-129 and microRNA-130 regulate VEGFR-2 expression in sensory and motor neurons during development.
Glaesel and colleagues describe the expression of VEGFR-2 in rat dorsal root ganglia and spinal cord and the involvement of microRNA-129 and microRNA-130 in the VEGFR-2 regulation.
The manuscript is generally well written, the objective is clear and valid interpretations.
Major points:
1) The authors show a western blot analysis to detect the expression of the VEGFR-2 protein in DRG and motor neurons. Adding a semiquantitative result on the analysis of more samples (main±DS) should improve the quality of the study.
2) The authors did not specify the sex of rats used in the experiments. If only one sex is included in the study, the title should specify the sex of animals. Moreover, in material and methods, authors should justify the reasons for any exclusion of females or males. In discussion, authors should further discuss the implications of the lack of analysis in both males and females on the interpretation of the results.
Minor points
1) Introduction line 89. The meaning of this sentence is not clear “Thereafter the expression of miR-129-5p and miR-130a-3p in these neurons was checked by using in situ hybridization and Western Blot”.
2) Results Paragraph 2.4, 2.5 and 2.6: The authors should indicate in the text the mean±DS values of mRNA and miRNA expression obtained by RT-qPCR
Reviewer 2 Report
Comments to Authors:
The subject addressed by the authors is of potential interest and the possible role of miR-129-5p and -130a-3p is worthy of investigation.
The authors show age-dependent miRNA expression in DRG neurons and spinal cord motor neurons in line with what already described in mature Purkinje cells of the cerebellum.
However, the authors fail to demonstrate a direct/indirect targeting of VEGFR by miR-129-5p and -130a-3p and the data shown are not sufficient to support any possible mechanistic model suggesting miR-129-5p and -130a-3p functional role in the modulation of VEGFR expression. In particular, the data shown in Figure 3 and Figure 4 only address the possible effect of miR-129-5p and miR-130a-3p, respectively, by qRT-PCR, whereas no experiments investigating the effect of miRNA-mediated regulation of VEGR protein levels are provided.
Moreover, it is not possible to infer any conclusion about a potential regulation of VEGFR by miR-129-5p or miR-130-3p in this cellualr context only on the basis of a minimal variation observed at the mRNA level, if that variation is below the 50 % of its expression level (due to the intrinsic limitation of the technique).
Therefore, additional experimental data and major revision is strongly recommended
Minor Comments:
-The authors are encouraged to proofread the manuscript and improve the quality of the English writing.
-line 93-99: This part of the manuscript represents a summary of what described in the following paragraphs of the Results section and should be actually moved to the Discussion section, if necessary.